# A bioinspired multilegged soft millirobot that functions in both dry and wet conditions

Haojian Lu [1], Mei Zhang[1], Yuanyuan Yang[1], Qiang Huang[2], Toshio Fukuda[2], Zuankai Wang [1,3] & Yajing Shen[1,3,4]

Developing untethered millirobots that can adapt to harsh environments with high locomotion efficiency is of interest for emerging applications in various industrial and biomedical settings. Despite recent success in exploiting soft materials to impart sophisticated functions which are not available in conventional rigid robotics, it remains challenging to achieve superior performances in both wet and dry conditions. Inspired by the flexible, soft, and elastic leg/foot structures of many living organisms, here we report an untethered soft millirobot decorated with multiple tapered soft feet architecture. Such robot design yields superior adaptivity to various harsh environments with ultrafast locomotion speed (>40 limb length/s), ultra-strong carrying capacity (>100 own weight), and excellent obstacle-crossing ability (stand up 90° and across obstacle >10 body height). Our work represents an important advance in the emerging area of bio-inspired robotics and will find a wide spectrum of applications.

[1] Department of Mechanical and Biomedical Engineering, City University of Hong Kong, Hong Kong, China. [2] Bejing Advanced Innovation Center for Intelligent Robots and Systems, Beijing Institute of Technology, Beijing, China. [3] Shenzhen Research Institute of City University of Hong Kong, Shenzhen 518057, China. [4] Department of Biomedical Engineering, City University of Hong Kong, Hong Kong, China. These authors contributed equally: Haojian Lu, Mei Zhang. Correspondence and requests for materials should be addressed to Z.W. (email: zuanwang@cityu.edu.hk) or to Y.S. (email: yajishen@cityu.edu.hk)

The fusion of soft materials with conventional robotics where rigid structures are dominantly implemented has sparked a wave of vigor and excitement in robotics science and engineering[1–3]. Indeed, owing to their adaptability to sophisticated terrain and safety for human interaction, the introduction of soft materials offers promise to overcome many obstacles inherent in conventional robots[4–13]. Recently, various soft micro- or milli-robots have been proposed based on different actuation mechanisms. In particular, owing to its inherent advantage in remote-control, magnetic guidance-based soft robot has received growing attention as an exemplifier by the helical magnetic robot[14–16] that can swim at low Reynolds number environment like bacterial/sperm and the film robot[13] that can realize multi-locomotion by folding and expanding its body. Although these strategies pave important foundation for in vivo bio-applications ranging from enhanced imaging to targeted drug delivery, the utility of soft structures also poses potential challenges such as relatively weak body supporting[17,18], low efficient locomotion[19–21], small carrying-load capacity[22–24], as well as limited obstacle-crossing ability[25,26]. These challenges become further severe at harsh environments where high humidity or liquid medium are involved. In such conditions, the performances of robots are dramatically compromised.

Legs and/or feet are commonly found in many living animals, including both land animals (e.g., ant, dog, cheetah, etc.) and ocean animals (e.g., starfish, octopus, etc.), after billions of years' evolution. The legs could lift the animal's body from ground in demand manner, leading to smaller body friction to ground, higher degrees of freedom in locomotion, less energy cost, and enhanced obstacle-crossing ability. Thus, legged animals usually demonstrate great adaptability to complex terrain, and can probably access virtually 100% of earth's land surface.

Inspired by these interesting structures in nature, in this work we present a new untethered milliscale (height ~1 mm) soft robot decorated with tapered feet structures to overcome existing challenges inherent in conventional soft robots. Under the trigger of external magnetic field, our robot can achieve a combined discontinuous and continuous locomotion, and the soft leg's motion is reminiscent of human's walking. In addition, the theoretical models are also built to elucidate how the tapered feet regulate robot's locomotion. Our robot demonstrates many superior functionalities in both wet and dry conditions such as excellent locomotion and deformability, heavy carrying capability, efficient locomotion ability, and over-obstacle ability. The seamless integration of bio-inspired multiple tapered feet structure with elastic and soft materials provides a general and powerful construction concept for the development of new robots and will find a wide spectrum of applications.

## Results

**Design of the multi-legged soft millirobot.** On the basis of extensive review of structural topographies of hundreds of legged animals, we find that the length of their legs ($L$) is normally 1~2 times larger than that of their foot-to-foot spacing ($S$). We hypothesis that it's a compromise between efficient locomotion and body supporting, since a large $L/S$ is preferential for locomotion, but inferior to body supporting; to contrary, a small $L/S$ is preferential for body supporting, but limited in locomotion. Interestingly, for animals with soft legs or feet such as starfish, centipede, and pillworm, the $L/S$ is typically approximately close to unity to make up the weakness of soft legs in body supporting (Supplementary Methods and Supplementary Figure 1). Herein, we chose $L/S$ as ~1:1 to balance the supporting and locomotion for the robot design.

Our robot is fabricated using a modified magnetic particle-assisted molding approach[27] (Fig. 1, Supplementary Methods and Supplementary Figure 2). Briefly, we first prepare a mixture

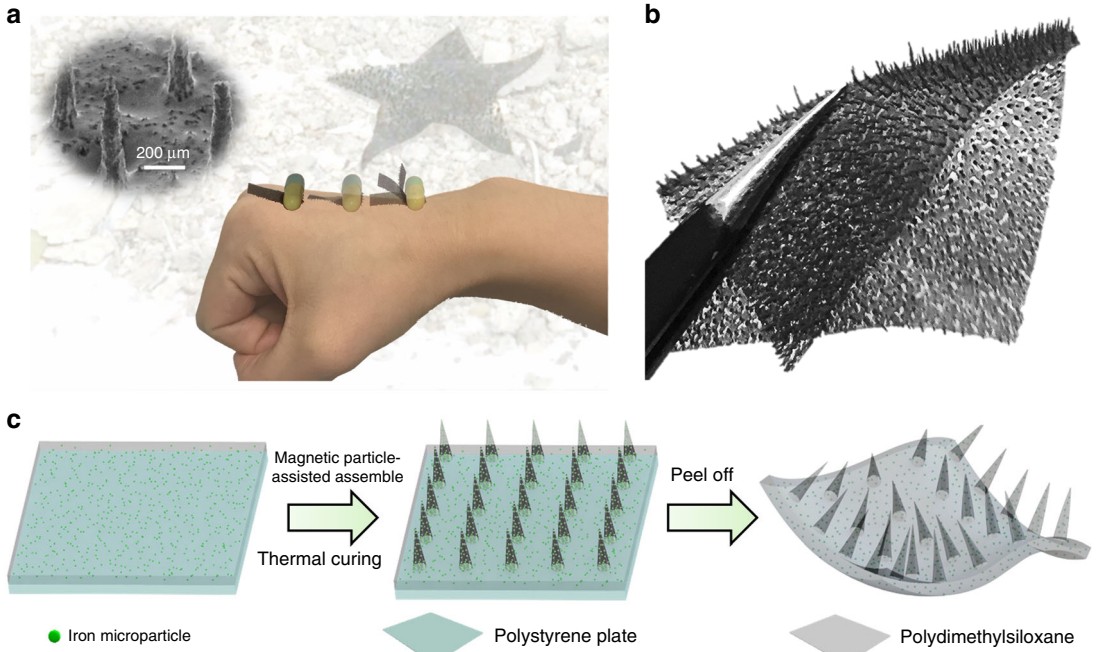

**Fig. 1** Multi-legged millirobot with flexible tapered feet. **a** Optical image of the soft robot with flexible tapered feet structure that exhibits superior adaptability, fast speed, heavy carrying–load ability, and excellent over-obstacle ability. **b** Optical image of the as-developed soft robot decorated with tapered feet. The thickness of body, foot length, and foot-to-foot spacing of robot is ~150 μm, ~650 μm, and ~600 μm, respectively. The tip angle of the tapered feet is ~15°, and the Yong's modulus of the as-fabricated soft body is measured to be ~2 MPa. **c** Schematic of robot fabrication process using a modified magnetic particle-assisted molding approach. Mixture containing polydimethylsiloxane (PDMS) pre-polymer and iron microparticles are first deposited on a polystyrene plate via spinning coating. Then, a magnet bar is put underneath to induce the formation of tapered feet under the guidance of magnetic field. After a solidification in a convection oven for thermal curing for 1 h at 80 °C, the as-fabricated film is peeled off from the plate and cut into the rectangular shape

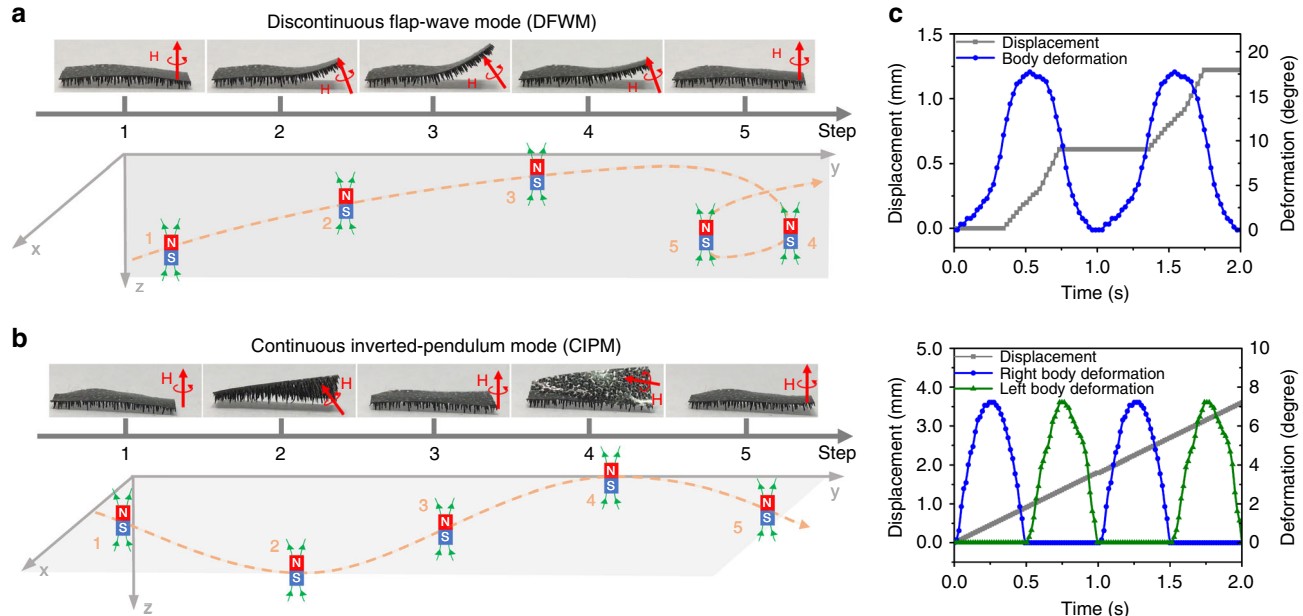

**Fig. 2** Two different locomotion modes. **a** Discontinuous flap-wave (DFW) locomotion under the application of an "O" shape magnetic field in the "y–z" plane, under which the robot shows a "stick-slip" locomotion. As the magnetic bar is moved upper and forward, the robot raises its body corresponding to the front feet alignment to the magnetic flux. At the same time, the robot moves forward one step under the pulling force along the y axis. After the external magnetic field is off, all the robot's feet are in touch with the ground again. Such a flap-wave locomotion is similar to a typical "stick-slip" movement widely used in piezo actuation at micro/nano scale. **b** Continuous inverted-pendulum (CIP) under the driving of an "S" shape magnetic field on "x–y" plane, under which the robot shows a "bipolar" locomotion-like human walking. The magnetic bar is programmed to move to the left and right directions alternately, meanwhile maintaining a forward movement. In response to the magnetic flux, the robot exhibits a continuous locomotion-like human walking, characterized by the alternate rise-up and continuous forward motion. **c** The robot's step size at the CIP mode (277% feet length) is around three times longer than that at the DFW mode (94% feet length) in each gait cycle. It indicates that the CIP locomotion has a higher locomotion efficiency than that of the DFW, mainly because the robot can switch twice in one gait cycle continuously at the CIP mode, whereas only once at the DFW mode

containing polydimethylsiloxane (PDMS), hexane, and magnetic particles. To induce the formation of tapered feet structures on our robot, an external magnetic field is applied during the solidification process. After peeling from the underlying substrate, we obtain a robot with the length of 17 mm, the width of 7 mm and the thickness of 150 μm (Fig. 1b). The $L$ and $S$ of the as-fabricated robot is ~650 and ~600 μm, corresponding to a $L/S$ ratio approximately close to unity. Such a value is also consistent with the signature of most of legged animals (Supplementary Figure 1). Since the Young modulus of the as-fabricated soft body is measured to be ~2 MPa, we refer to our robot as soft multi-legged millirobot.

**Robot driving and locomotion model**. We regulate the robot's dynamic locomotion in a remote fashion by employing a permanent magnet (Fig. 2). When a magnetic field is applied, both magnetic torque and pulling force will be generated[28,29]. Thus, the tapered feet are deformed and align with the direction of magnetic flux, and the robot moves forward displaying various postures, in response to the combined action of the magnetic. Here, the maximum magnetic torque applied to a single flexible tapered foot is ~0.4 nN·m according to both theoretical analysis and experimental measurement (Supplementary Methods and Supplementary Figure 3). In contrast, there is no any marked movement for the robot without the design of feet structure, even when the magnetic field is applied up to 200 mT (Supplementary Figure 4). Moreover, on both wet and dry surfaces, the friction force is at least 40 folds smaller than that of its counterpart (see Supplementary Methods and Supplementary Figure 5). This significant reduction in the friction force results from the reduced

contact area between the robot and ground, which is ~5000 folds smaller than that of its counterpart. Notably, all these results are achieved without the loss of structural flexibility. More details about the magnetic mechanism for robot driving can be found from Supplementary Methods and Supplementary Figure 3.

Under the trigger of external magnetic field, our robot can achieve a combined multiple locomotion, i.e., the combination of discontinuous flap-wave (DFW) locomotion and a continuous inverted-pendulum (CIP) locomotion. A DFW locomotion is generated by the application of magnet with an "O" trajectory in the y–z plane (Fig. 2a and Supplementary Movie 1). Initially, when the magnet is located underneath the robot, the feet of robot stick on the ground. As the magnetic bar is moved upper and forward, the front feet rise in alignment with the magnetic flux (Fig. 2a) and the robot moves forward step by step. After the external magnetic field is off, the robot's feet are in touch with the ground again. Such a flap-wave locomotion is similar to the typical "stick-slip" movement widely used in piezo actuation at micro/nano scale. In contrast, the use of magnet with a "S" trajectory in the x–y plane results in a CIP locomotion (Fig. 2b and Supplementary Movie 1). In this kind of locomotion, the magnetic bar is programmed to move to the left and right directions alternately, meanwhile maintaining a forward movement. In response to the magnetic flux, the robot exhibits a continuous locomotion-like human walking, characterized by the alternate rise-up and continuous forward motion.

Our experimental results further demonstrate that the CIP locomotion has a higher locomotion efficiency than that of the DFW, though the latter usually gives a better obstacle-crossing ability. In terms of the stride frequency, the robot can switch

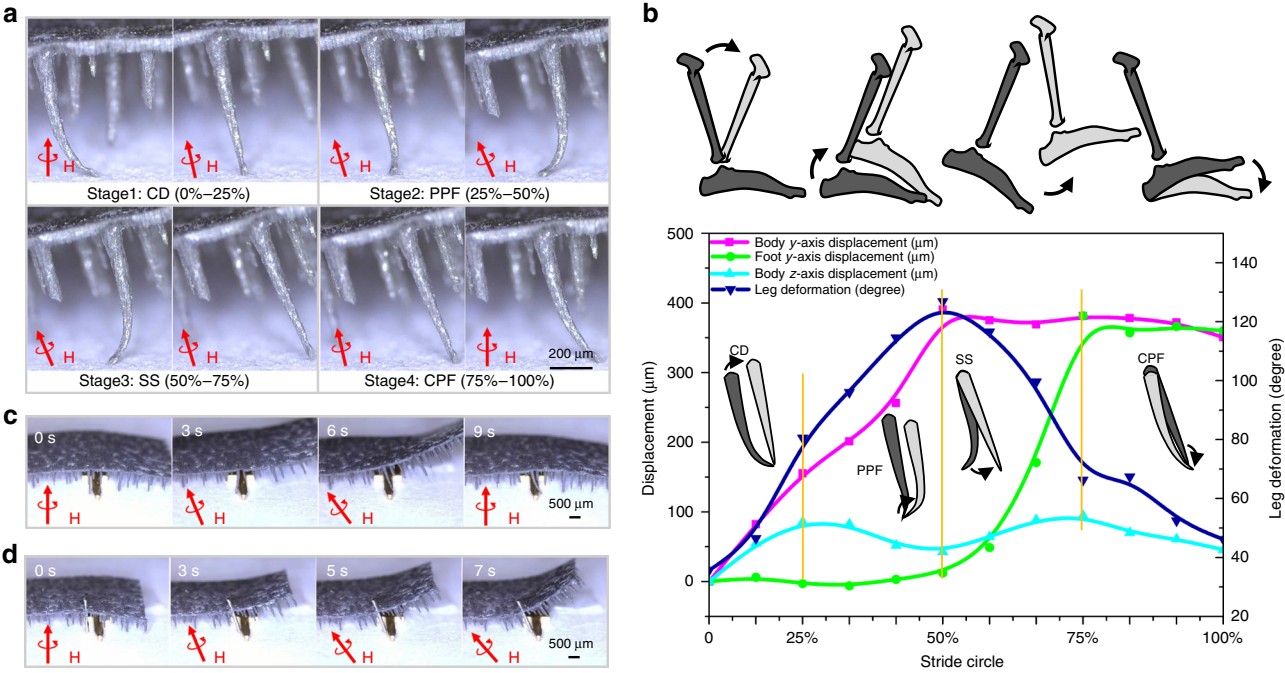

**Fig. 3** Individual foot's performance in one gait cycle. **a** Snapshots of an individual foot's deformation in one gait cycle, which can be divided into four typical stages: controlled dorsiflexion (CD), power plantar flexion (PPF), swing state (SS), and controlled plantar flexion (CPF). **b** Locomotion curves, including the robot body's displacement along y axis (pink square line), the foot's displacement along y axis (green circle line), the robot body's displacement along z axis (light blue regular triangle line), and the foot's deformation (dark blue inverted triangle line). The robot can maintain a smooth transition among four different stages in each gait cycle, which is reminiscent of that of human being and overcome the singular point of the traditional robot with rigid legs. **c**, **d** The robot with flexible feet (Fig. 3c) exhibits superiority in obstacle crossing relative to the counterpart with the rigid feet (Fig. 3d). When the robot is exposed to an obstacle (height 160 μm × width 1 mm) on the way forward, it can stride via quick release of the stored elastic energy enabled by the feet deformation. In contrast, the robot decorated with rigid feet is always stuck by the obstacle, irrespective of the direction of the magnetic flux applied

twice in one gait cycle continuously at the CIP mode, whereas only once at the DFW mode. Moreover, as shown in Fig. 2c, under the same magnetic strength (200 mT), the step size (277% foot length) in one gait cycle of CIP locomotion is almost three-fold longer than that of the DFW mode. In addition, the locomotion efficiency at the CIP mode is twice higher than that of the human's walking. More details about the modeling and simulation of DFW and CIP locomotion can be found from the supplementary materials (see Supplementary Methods and Supplementary Figure 6).

To reveal the basic mechanisms underpinning the intriguing locomotion manifested in our soft robot, we further analyze the continuous motion of individual foot in one gait cycle. We record the robot's posture using high-speed camera (Fig. 3a), and plot the robot body's displacement along y axis (pink square line), the foot's displacement along y axis (green circle line), the robot body's displacement along z axis (light blue regular triangle line), and the foot's deformation (dark blue inverted triangle line), respectively (Fig. 3b). It is clear that similar to that of human being, the robot's gait can be divided into four stages based on interplay between the robot's displacement and the stride circle: controlled dorsiflexion (CD, 25%), power plantar flexion (PPF, 25–50%), swing state (SS, 50–75%) and controlled plantar flexion (CPF, 75–100%) (Fig. 3b, and Supplementary Methods).

The flexible, soft feet can store (PPF and CPF stages) and release elastic energy (CD and SS stages) alternatively in response to external magnetic field in each gait cycle. We find that the value of the stored/released elastic energy is approximately two orders of magnitude higher than that of the gravity energy (see Supplementary Methods, Supplementary Figure 6, and Supplementary Movie 2). This alternative energy storage-release

process not only reduces energy cost and increases the locomotion efficiency, but also serves as a damper to improve the stability (see Supplementary Methods). Indeed, as reflected in the plot of the robot body's displacement along z axis in Fig. 3b (light blue line), the robot can maintain a smooth transition among four different stages in each gait cycle, overcoming the singular point of the traditional robot with rigid legs[30]. Furthermore, compared with the rigid ones, the soft feet also give rise to superior over-obstacle ability, owing to the remarkable deformable ability and quick elastic energy release mechanism. As demonstrated in Fig. 3c, when the robot is exposed to an obstacle (height 160 μm × width 1 mm), it can stride via quick release of the stored elastic energy. In contrast, the robot decorated with rigid feet is always stuck by the obstacle, irrespective of the direction of the magnetic flux applied (Fig. 3c and Supplementary Movie 3).

**Robot motion at harsh environment**. To demonstrate the versatility of our robot, we further perform the operation of robots in several harsh environments, including wet slippery surfaces, heavy loading, and high sloppy obstacles. Previously it has been recognized that locomotion on slippery surfaces is very challenging due to the presence of sticky water layer[31]. In our design, the tapered foot leads to small contact area (78 $\mu m^2$/foot out of 3.6 × $10^5$ $\mu m^2$) with the underlying substrate. Moreover, the roughness surface[32] with intrinsic hydrophobic property (contact angle ~115°) further makes the surface superhydrophobic (Supplementary Movie 4, Supplementary Figure 7). As a result, on both dry and wet environments, the friction forces between the robot and ground are reduced by >40 times (see Supplementary

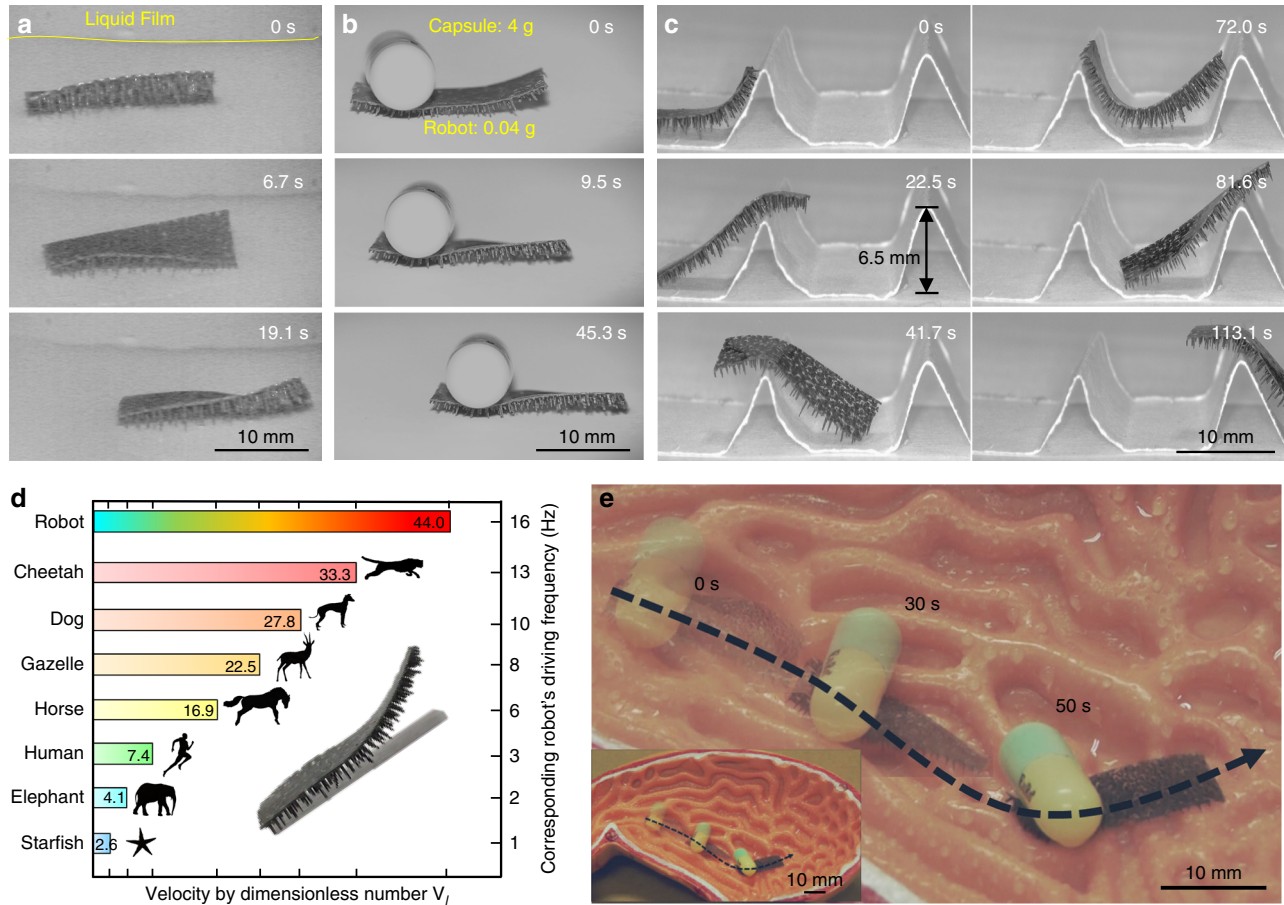

**Fig. 4** Demonstration of robot locomotion at harsh environment. **a** Locomotion on wet surface with liquid film. The robot can move with an average speed of 0.5 mm/s on wet surface under a drive frequency of 1 Hz owing to its small contact area to underlying substrate and hydrophobic tapered feet. **b** Robot locomotion with a loading 100 times of its own weight. In this condition, the robot can move 8 mm in 45 s. **c** Cross a steep obstacle with height ~10 times higher of its own leg. The robot's posture can be easily adjusted to climb obstacles as a result of the discontinuous flap-wave mode and continuous inverted-pendulum motion locomotion mode. **d** Comparison of the normalized speed between the soft robot and other animals. A dimensionless number $V_l$, i.e., the locomotion distance in one second relative to the lower limb (leg or foot) length, is defined to quantify the locomotion efficiency of various living organisms. The maximum velocities of human and cheetah are 10.4 m/s and 33.3 m/s, respectively, corresponding to $V_l$ 7.4 and 33.3, respectively. In contrast, the soft robot can easily reach the maximum speed of human and cheetah at a swing frequency 3 Hz and 13 Hz, respectively. **e** Demonstration of drug transport in a stomach model under wet environment. The complex stomach internal structure is 1.5–6.8 mm in depth and 2.4–6.2 mm in width (2–10 orders of magnitude lager than the leg length). The robot can move 32 mm in 50 s at such a harsh in vivo simulated environment, meanwhile carrying a medical tablet (~91.4 mg) that is twice heavier than itself

Methods and Supplementary Figure 5). As shown in Fig. 4a, the robot can move 10 mm with an average speed of 0.5 mm/s on the wet surface under the drive frequency 1 Hz (Supplementary Movie 4). Also, the tapered feet ensure a point contact to ground even when heavy loads are applied. As shown in Fig. 4b, a robot of 39.4 mg can take a capsule with weight up to 3980.6 mg (a capsule filled by Pb beads), which is >100 times heavier than itself. Moreover, under the drive frequency of 1 Hz, the robot can move 8 mm in 45 s (Supplementary Movie 5). Such a carrying capability is comparable to ant, i.e., one of the strongest Hercules in nature, and stronger than most animals. Moreover, the flexible feet serve as both hanger and damper in locomotion, leading to mutual enhancement in stability and wall climbing ability. As a demonstration, we set two obstacles with a height of 6.5 mm and the slope angle of 60° (Fig. 4c). The robot can pass both obstacles within 113 s. More impressively, our robot is able to rise its body up to 90° and cross the steep obstacle which is 10 times higher than the length of its leg (Fig. 1 and Supplementary Movie 6).

Furthermore, we demonstrate that our robot can achieve an ultra-high locomotion speed. To quantify the locomotion

efficiency of various lives, we define a dimensionless number $V_l$, i.e., the locomotion distance in one second relative to the lower limb (leg or foot) length. As illustrated in Fig. 4d, the maximum velocities of human and cheetah are 10.4 m/s and 33.3 m/s, respectively, corresponding to $V_l$ 7.4 and 33.3, respectively. In contrast, our robot can easily reach the maximum speed of human and cheetah at a swing frequency 3 Hz and 13 Hz, respectively (Fig. 4d and Supplementary Movie 7). Notably, such swing frequencies are much smaller than the natural frequency of the body or leg of our robot (~3.4 kHz and ~27.9 Hz, respectively, see Supplementary Methods and Supplementary Figure 8). As the experimental data show in Fig. 4d, the $V_l$ of our robot can reach 44 under a swing frequency 16 Hz, which is 30% faster than that of cheetah.

To further exhibit the potential applications in in vivo biomedical environment, we demonstrate the locomotion of our robot on a human stomach-like structure (isolated, wet surface) (Fig. 4e). We choose the complex stomach internal structure with 1.5–6.8 mm in depth and 2.4–6.2 mm in width (2–10 orders of magnitude lager than the leg length). To demonstrate the carrying

ability, we adhere a medical tablet (~91.4 mg) on the robot body, which is twice times heavier of its own weight. As the result shows, our robot can move 32 mm in 50 s at such harsh in vivo simulated environment by untethered control (Supplementary Movie 8).

Lastly, in addition to the rectangular robot demonstrated above, the thin sheet decorated with multiple tapered feed can be also trimmed to other shape to mimic the animals in nature, such as starfish. As shown in Supplementary Methods, Supplementary Figure 9, and Supplementary Movie 9, the constructed starfish robot can overcome the obstacles and locomote efficiently at a simulated live harsh environment. We believe the superiority of untethered magnetic control[33–35] and the multiple feet sheet structure could promise tremendous opportunities for robot design in the future.

## Discussion

Legs and/or feet play an important role in the locomotion of animals by offering effective body supporting, higher motion agility and better obstacle over cross ability, etc, and allows the animal to probably access virtually 100% of earth's land surface. After reviewing hundreds of legged animals, we find that $L/S$ is ~1 to balance the body supporting and effective locomotion of soft animals. Inspired by these facts in nature, we design a multi-legged soft millirobot by a modified magnetic particle-assisted molding approach. The legs can lift the body from ground effectively and reduce the friction force 40 folds at least at dry condition. Notably, benefiting from the legged structure and the hydrophobic material, our robot can locomote on the wet surface efficiently.

The locomotion of our robot is regulated in a remote fashion by magnetic field dynamically. Different to the conventional magnetic driven approach, we employ both magnetic torque and pulling force as sources to drive the robot, leading to better adaptability to various surroundings. Under the trigger of external magnetic field, our robot can achieve a combined multiple locomotion, i.e., the combination of DFW locomotion and a CIP locomotion. In addition, the flexible legs can store and release elastic energy during locomotion, which not only reduces energy cost, but also serves as a damper to improve the stability and increase over-obstacle ability. More interestingly, the leg's loco-motion in one gait cycle exhibits a high similarity to that of human walking.

As demonstrated in the experiment, the multi-legged design offers the soft millirobot many advances, including excellent locomotion and deformability, heavy carrying capability, efficient locomotion ability, over-obstacle ability, and high adaptability to both dry and wet environment. The concept of integrating mul-tiple tapered feet with soft materials provides a general and powerful strategy for soft robot development, and it would benefit a wide range of fields such as untethered manipulation in poorly accessible space, movable laminated sensing, and in vivo medical transportation.

## Data availability

The data that support the findings of this study are available from the corresponding author on reasonable request.

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

## Acknowledgements

This work was supported in part by Shenzhen (China) Basic Research Project (No. JCYJ20160329150236426), Research Grants Council of the Hong Kong Special Administrative Region (No. C1018-17G, No. 11218417), and National Science Foundation of China (No. 61773326).

## Author contributions

H.L. did the experiment and analyzed the data. M.Z. fabricated the soft robot. Y.Y. studied the friction property of the robot. Q.H. and T.F. provided valuable advices in robot locomotion control. Z.W. and Y.S. initialized the idea, designed the experiment, and analyzed the data.

## Additional information

**Competing interests:** The authors declare no competing interests.

