## [Peer Review File · Nature Communications]

Reviewer #1 (Remarks to the Author):

The authors provided extensive replies to the reviewers' comments and the revised paper has improved.

However, most of the concerns about the work and its possible publication in a Nature journal are still standing.

While the work can be valuable in some aspects, the paper does not highlight the scientific contribution and does not always support the (over)statements.

Reviewer #2 (Remarks to the Author):

I have read through the manuscript, response to the reviewers, and the supplemental materials. I appreciate the authors' thorough responses to all of my comments. Based on the significant changes, I believe the manuscript is acceptable in Nature Communications. However, I need to admit that I did not find adequate time to review the submission in the level of detail required to write a point-by-point review. My overall assessment is that the manuscript presents valuable results that should be published.

Reviewer #3 (Remarks to the Author):

This manuscript reports a demonstration of a magnetic millirobot based on magnetic pillars that are formed through self-assembly on one side of a PDMS film. This film is placed with the pillars pointing downward on top of the surface (including complex and wet terrains). Through rotation and translation of a permanent magnet below the stage for walking, torques and forces are generated that result in net forward motion of the millirobot, where the pillars serve as feet. Two primary modes of walking are presented, discontinuous flap wave, and continuous inverted pendulum. Significant modeling results are also presented to explain the mechanism of locomotion. The authors

have made substantial improvements in the revised manuscript, and I recommend it for publication after addressing the following issue:

This manuscript places much more emphasis on the mechanics rather than the magnetics aspects of these millirobots, which is acceptable. The Supplementary Methods Section 4, Magnetic Mechanism Analysis of Single Tapered Foot, provides a theoretical model, but it does not “quantitatively analyze the magnetic torque of tapered feet.” Such quantitative analysis would typically refer to an experimental measurement. Furthermore, the authors assume the particles form “single chains,” which likely is not the case and cannot be assumed without structural verification. The torques on thicker chains would be weaker and would depend on the width of the chain. The authors refer to Extended Data Fig. 6a, which is a cartoon, not experimental measurements. This claim of quantitative analysis of magnetic torques should be removed or modified to address this issue, or additional measurements and/or modeling should be provided to provide quantitative results.

Reviewer #1 (Remarks to the Author):

The authors provided extensive replies to the reviewers' comments and the revised paper has improved. However, most of the concerns about the work and its possible publication in a Nature journal are still standing. While the work can be valuable in some aspects, the paper does not highlight the scientific contribution and does not always support the (over)statements.

Response: We are pleased that the referee acknowledged that the quality of our revised manuscript has improved. We would like to highlight the novelty and contributions of this work as below.

To our best knowledge, this is the first study that reports the development of an untethered soft millirobot decorated with multiple tapered feet structure that yields superior performances. As discussed in our manuscript, our robot exhibits several superiorities in locomotion, such as high adaptivity to harsh environments (Supplementary Video 4), heavy loading ability (Supplementary Video 5), obstacle overcoming ability (Supplementary Video 6), fast locomotion speed (Supplementary Video 7), etc, and offers a wide range of applications (Supplementary Video 8). Moreover, owing to the unique multi-legged structure, our robot can be driven by the combined action of magnetic torque and force (Extended Data Fig. 3).

We also perform systematical theoretical analysis to elucidate the driving mechanisms underlying our soft robot, including the magnetic torque analysis (Supplementary Methods Section 3 and 4), legged soft robot's locomotion analysis (Supplementary Methods Section 6 and 7), energy variation analysis (Supplementary Methods Section 8 and 9) and natural frequency analysis (Supplementary Methods Section 10). We also include a new theoretical model to elucidate the magnetic driven mechanism and verified it quantitatively by adding a new experiment (see Supplementary Methods Section 4 and Extended Data Fig. 3).

We emphasize that our work is novel, significant and appealing. The introduction of the multiple tapered feet concept to the soft robot design dramatically extends our capability in the development of robust robots that exhibits functionalities otherwise impossible. We expect you are now satisfied with our revision and consider our publication in *Nature Communications*.

Reviewer #2 (Remarks to the Author):

I have read through the manuscript, response to the reviewers, and the supplemental materials. I appreciate the authors' thorough responses to all of my comments. Based on the significant changes, I believe the manuscript is acceptable in Nature Communications. However, I need to admit that I did not find adequate time to review the submission in the level of detail required to write a point-by-point review. My overall assessment is that the manuscript presents valuable results that should be published.

Response: We thank the referee very much for recommending our manuscript for publication in *Nature Communications*.

Reviewer #3 (Remarks to the Author):

This manuscript reports a demonstration of a magnetic millirobot based on magnetic pillars

that are formed through self-assembly on one side of a PDMS film. This film is placed with the pillars pointing downward on top of the surface (including complex and wet terrains). Through rotation and translation of a permanent magnet below the stage for walking, torques and forces are generated that result in net forward motion of the millirobot, where the pillars serve as feet. Two primary modes of walking are presented, discontinuous flap wave, and continuous inverted pendulum. Significant modeling results are also presented to explain the mechanism of locomotion. The authors have made substantial improvements in the revised manuscript, and I recommend it for publication after addressing the following issue:

This manuscript places much more emphasis on the mechanics rather than the magnetics aspects of these millirobots, which is acceptable. The Supplementary Methods Section 4, Magnetic Mechanism Analysis of Single Tapered Foot, provides a theoretical model, but it does not “quantitatively analyze the magnetic torque of tapered feet.” Such quantitative analysis would typically refer to an experimental measurement. Furthermore, the authors assume the particles form “single chains,” which likely is not the case and cannot be assumed without structural verification. The torques on thicker chains would be weaker and would depend on the width of the chain. The authors refer to Extended Data Fig. 6a, which is a cartoon, not experimental measurements. This claim of quantitative analysis of magnetic torques should be removed or modified to address this issue, or additional measurements and/or modeling should be provided to provide quantitative results.

Response: We thank the referee very much for his/her constructive and insightful comments on providing quantitative results to analyze the magnetic torque of the tapered feet.

To substantiate our conclusions, we performed additional measurements suggested by the referee. Briefly, taking a single flexible tapered foot as a unit, the magnetic torque \vec{T}_{sm} applied to the tapered foot can be calculated regardless of the foot’s structure (“single chains” or “thick chains”). As illustrated in **Fig. R1a**, the magnetic torque applied to the tapered foot can be calculated as [R1]:

$$\vec{T}_{sm} = \int_{V_{sm}} d\vec{T}dV = \int_{V_{sm}} \left((M_y B_z - M_z B_y) \vec{i} + (M_z B_x - M_x B_z) \vec{j} + (M_x B_y - M_y B_x) \vec{k} \right) dV \quad (\text{Eq1})$$

where \vec{M} is the magnetization of the tapered foot, V_{sm} is the volume of the single flexible tapered foot and $\vec{B}(x, y, z)$ is the magnetic flux intensity. Suppose the angle between \vec{M} and z axis is α , when the permanent magnet located below the robot moves forward along y axis, the magnetic torque along x , y and z axis can be represented as:

$$\left\{ \begin{array}{l} \vec{T}_{smx} = B_z \sin \alpha \int_{V_{sm}} M dV - B_y \cos \alpha \int_{V_{sm}} M dV \\ \vec{T}_{smy} = B_x \cos \alpha \int_{V_{sm}} M dV \\ \vec{T}_{smz} = -B_x \sin \alpha \int_{V_{sm}} M dV \end{array} \right. \quad (\text{Eq2})$$

where

$$B_x = \frac{\mu_0 J_s}{8\pi} \left[-\Gamma(0.5ma - x, 0.5mb - y, 0.5mc + z) - \Gamma(0.5ma - x, 0.5mb + y, 0.5mc + z) \right. \\ \left. + \Gamma(0.5ma + x, 0.5mb - y, 0.5mc + z) + \Gamma(0.5ma + x, 0.5mb + y, 0.5mc + z) \right] \quad (\text{Eq3})$$

$$B_y = \frac{\mu_0 J_s}{8\pi} \left[-\Gamma(0.5mb - y, 0.5ma - x, 0.5mc + z) - \Gamma(0.5mb - y, 0.5ma + x, 0.5mc + z) \right. \\ \left. + \Gamma(0.5mb + y, 0.5ma - x, 0.5mc + z) + \Gamma(0.5mb + y, 0.5ma + x, 0.5mc + z) \right] \quad (\text{Eq4})$$

$$B_z = \frac{\mu_0 J_s}{4\pi} \left[-\Psi(0.5mb - y, 0.5ma - x, 0.5mc + z) - \Psi(0.5mb + y, 0.5ma - x, 0.5mc + z) \right. \\ - \Psi(0.5ma - x, 0.5mb - y, 0.5mc + z) - \Psi(0.5ma + x, 0.5mb - y, 0.5mc + z) \\ - \Psi(0.5mb - y, 0.5ma + x, 0.5mc + z) - \Psi(0.5mb + y, 0.5ma + x, 0.5mc + z) \\ \left. - \Psi(0.5ma - x, 0.5mb + y, 0.5mc + z) - \Psi(0.5ma + x, 0.5mb + y, 0.5mc + z) \right] \quad (\text{Eq5})$$

$$\Gamma(\gamma_1, \gamma_2, \gamma_3) = \ln \left. \frac{\sqrt{\gamma_1^2 + \gamma_2^2 + (\gamma_3 - z_0)^2} - \gamma_2}{\sqrt{\gamma_1^2 + \gamma_2^2 + (\gamma_3 - z_0)^2} + \gamma_2} \right|_{z_0 = mc}^{z_0 = 0} \quad (\text{Eq6})$$

$$\Psi(\psi_1, \psi_2, \psi_3) = \arctan \left[\frac{\psi_1(\psi_3 - z_0)}{\psi_2 \sqrt{\psi_1^2 + \psi_2^2 + (\psi_3 - z_0)^2}} \right] \left. \right|_{z_0 = 0}^{z_0 = mc} \quad (\text{Eq7})$$

During the experiment, the permanent magnet is put underneath the robot with a distance of 20 mm, and then activated in the y direction to drive the robot forward. Here, the permeability of vacuum μ_0 is $4\pi \times 10^{-7}$ H/m, the surface density of magnetizing current J_s is 3.1×10^4 A/m, the length, width and height of the rectangular permanent magnet are $ma=40$ mm, $mb=40$ mm, $mc=20$ mm, respectively. The magnetization of the tapered foot \vec{M} is measured experimentally by a vibrating sample magnetometer (Cryogenic Inc.) (**Fig. R1b**). By substituting these values into Eqs.2-7, the theoretical magnetic torque applied to the tapered foot can be successfully calculated, as indicated by the blue curve in **Fig. R1c**.

To further validate our theoretical model, we calculate the actual torque at various locomotion stages based on the leg's deformation by classical Euler-Bernoulli equation [R2]. Remarkably, as shown in **Fig. R1c**, the data predicted from our theoretical model are in good agreement with these obtained experimentally, suggesting the efficacy of our proposed model.

Please find Supplementary Methods Section 4 and Extended Data Figure 3 in the manuscript for more details.

Figure R1 | Magnetic torque analysis of a single foot. a, Schematic drawing of the single tapered foot. The single tapered foot is treated as a whole unit regardless of the structure. **b,** The magnetization of the single tapered foot under different applied magnetic field strength. As the magnetic field strength applied at the tapered feet increases from 0 mT to 200 mT, the magnetization of the tapered feet increases from 0.08 kA/m to 4.22 kA/m. **c,** The magnetic torque of the single tapered foot. The theoretical value of magnetic torque applied to the tapered foot agrees well with the experimental results, which verifies the model.

References:

[R1] Mahoney A W, Abbott J J. Five-degree-of-freedom manipulation of an untethered magnetic device in fluid using a single permanent magnet with application in stomach capsule endoscopy[J]. The International Journal of Robotics Research, 2016, 35(1-3): 129-147.
 [R2] Fertis D G. Nonlinear structural engineering[M]. Springer-Verlag Berlin Heidelberg, 2006.

Reviewer #1 (Remarks to the Author):

The authors did not revise the paper significantly in this round. Most of the concerns about the work and its possible publication in a Nature journal are still standing. I confirm that, while the work can be valuable in some aspects, the paper does not highlight the scientific contribution and does not always support the (over)statements.

Reviewer #3 (Remarks to the Author):

The authors have appropriately responded to my concerns. I recommend this manuscript for publication in Nature Communications.

Reviewer #1 (Remarks to the Author):

The authors did not revise the paper significantly in this round. Most of the concerns about the work and its possible publication in a Nature journal are still standing. I confirm that, while the work can be valuable in some aspects, the paper does not highlight the scientific contribution and does not always support the (over)statements.

Response: We thank the referee very much for insightful comments. In this version, we have further stated the contribution and novelty of our work in the discussion session. Briefly:

Inspired by the leg structure of animals in nature, we design a multi-legged soft millirobot by a modified magnetic particle-assisted molding approach after reviewing hundreds of legged animals. The legs can lift the body of soft robot from ground effectively and reduce the friction force 40 folds at least at dry condition. Notably, benefiting from the legged structure and the hydrophobic material, our robot can locomote on the wet surface efficiently. To our best knowledge, this is the first study that reports the development of an untethered soft millirobot decorated with multiple tapered feet structure that yields superior performances.

Different to the conventional magnetic driven approach, we employ both magnetic torque and pulling force as sources to drive the robot, leading to better adaptability to various surroundings. In addition, the flexible legs can store and release elastic energy during locomotion, which not only reduces energy cost, but also serves as a damper to improve the stability and increase over-obstacle ability. More interestingly, the leg's locomotion in one gait cycle exhibits a high similarity to that of human walking. The multi-legged design offers the soft millirobot many advances, including excellent locomotion and deformability, heavy carrying capability, efficient locomotion ability, over-obstacle ability and high adaptability to both dry and wet environment.

Overall, we believe this work is a great progress in this field. The proposed robot design concept provides a general and powerful strategy for soft robot design and it will benefit a wide range of fields such as untethered manipulation in poorly accessible space, movable laminated sensing, and *in-vivo* medical transportation. We also believe that it will attract a wide interest of the readers of Nature Communications. We expect you are satisfied with our adaptation and consider our publication in Nature Communications.

Reviewer #2 (Remarks to the Author):

The authors have appropriately responded to my concerns. I recommend this manuscript for publication in Nature Communications.

Response: We thank the referee very much for accepting this manuscript for publication. We will do more work in this area to promote the legged soft robot.